# Use of Recycled Additive Materials to Promote Efficient Use of Resources While Acting as an Effective Toughness Modifier of Wood–Polymer Composites

**DOI:** 10.3390/polym16182549

**Published:** 2024-09-10

**Authors:** Luísa Rosenstock Völtz, Linn Berglund, Kristiina Oksman

**Affiliations:** 1Division of Materials Science, Department of Engineering Sciences and Mathematics, Luleå University of Technology, SE-97187 Luleå, Sweden; luisa.voltz@ltu.se (L.R.V.); linn.berglund@ltu.se (L.B.); 2Wallenberg Wood Science Center (WWSC), Luleå University of Technology, SE-97187 Luleå, Sweden; 3Department of Mechanical & Industrial Engineering (MIE), University of Toronto, Toronto, ON M5S 3G8, Canada

**Keywords:** wood–polymer composites, recycled modifiers, impact properties, fracture toughness, microtomography

## Abstract

Wood–polymer composites (WPCs) with polypropylene (PP) matrix suffer from low toughness, and fossil-based impact modifiers are used to improve their performance. Material substitution of virgin fossil-based materials and material recycling are key aspects of sustainable development and therefore recycled denim fabric, and elastomer were evaluated to replace the virgin elastomer modifier commonly used in commercial WPCs. Microtomography images showed that the extrusion process fibrillated the denim fabric into long, thin fibers that were well dispersed within the WPC, while the recycled elastomer was found close to the wood fibers, acting as a soft interphase between the wood fibers and PP. The fracture toughness (K_IC_) of the WPC with recycled denim fabric matched the commercial WPC which was 1.4 MPa m^1/2^ and improved the composite tensile strength by 18% and E-modulus by 54%. Recycled elastomer resulted in slightly lower K_IC,_ 1.1 MPa m^1/2^, as well as strength and modulus while increasing elongation and contributing to toughness. The results of this study showed that recycled materials can potentially be used to replace virgin fossil-based elastomeric modifiers in commercial WPCs, thereby reducing the CO_2_ footprint by 23% and contributing to more efficient use of resources.

## 1. Introduction

The addition of wood flour/fibers in thermoplastics has many benefits, such as reduced material cost, increased mechanical properties, and decreased weight compared with other fillers, and wood is derived from renewable resources [1,2]. Residual wood from industries, as well as recycled thermoplastics, has been employed to produce WPCs, making these composites partially or completely produced of recycled materials. However, the addition of wood flour or fibers into polypropylene results in a decrease in their impact strength and fracture toughness [1,2,3,4]. Few studies have investigated the impact and fracture properties of WPCs [3,5], the use of modifiers [1,4,6,7,8,9,10,11,12,13,14], or examples where recycled materials were used as impact modifiers in WPCs are challenging to find.

The most common strategy for improving the toughness of WPCs includes the addition of soft elastomeric modifiers to make the matrix softer or create a soft interface [1,9,12,15]. The fracture mechanism of soft inclusions is that these are more ductile than the polymer matrix, and act as stress concentrators, resulting in improved impact strength via crazing, shear yielding, microcracks, and/or cavitation [15,16,17,18]. It has been found that the presence of elastomers may form two structures: (1) elastomers are dispersed in the polymeric matrix as a third phase, and/or (2) elastomers encapsulating the rigid reinforcement, resulting in improved fiber–matrix adhesion [1,9,11]. Elastomers, such as ethylene–propylene–diene–monomer (EPDM) [1,9,10], styrene–ethylene–butylene–styrene (SEBS) [1,6], and polyolefin elastomers (POE) [8,11,12], have been used as impact and toughness modifiers for WPCs. For example, Oksman and Clemons [1] added 10 wt% of elastomers, EPDM, EPDM–g–maleic anhydride (MA), and SEBS–g–MA to polypropylene (PP)–wood flour composites, where the addition of EPDM decreased the tensile strength and modulus, but increased the notched Izod impact strength from 26 to 38 J/m, and EPDM–g–MA and SEBS–g–MA further increased the impact strength to 48 and 51 J/m, respectively. They suggested that the elastomers–g–MA formed a flexible interphase around the wood particles and matrix, resulting in a higher impact strength compared with EPDM alone.

Recently, tough, and long fibers have been studied to improve composite toughness and impact properties, these are often referred to as hybrid composites [13,14,19,20,21]. The fracture mechanism in these composites is the higher absorption energy capacity from debonding and pull-outs of long fibers resulting in better impact strength and toughness. However, its effectiveness depends on several factors, including the fiber properties, fiber aspect ratio, and the interface between the matrix and fibers [19,20]. It has been found that longer fibers in composites bridge a higher number of cracks, resisting crack opening and crack propagation, whereas short fibers act as flaws, reducing the aforementioned properties [19,21].

Várdai et al. have studied the use of long fibers in WPCs, such as poly(ethylene terephthalate) (PET) and poly(vinyl alcohol) (PVA) fibers in the WPCs [13,14]. In one of the studies, they added short PET fibers with a length of 4 mm up to 40 wt% for hybridization of WPCs with 20 wt% wood flour [13]. They showed that the addition of approx. 40 wt% PET fibers in the WPC resulted in the highest impact resistance when maleic anhydride grafted polypropylene (MAPP) was used as a coupling agent, the impact strength increased from 2 to 15 kJ/m^2^ and they suggested that the PET fibers hindered crack initiation and propagation, preventing catastrophic failure by increasing the plastic deformation of the matrix. In another study, the same authors used PVA fibers, with a length of 3 mm for hybridization of PP-wood flour composites (20 wt%) and showed that the addition of 20 wt% PVA fibers had a positive effect resulting in an impact resistance of approx. 25 kJ/m^2^ [14]. The results of these studies showed that polymer fibers are excellent impact modifiers for WPCs; however, the feeding of short polymer fibers can be challenging in the continuous extrusion process.

Although the use of elastomers as matrix modifiers in WPCs is an effective and well-known practice in industry the soft elastomer has a negative impact on strength and stiffness, and it will also increase the cost. In addition, virgin fossil-based materials are increasing competition for limited resources [22]. To provide a resource-efficient solution for fossil-based modifiers and reduce waste generation, targeting the United Nations Sustainable Development Goals (SDGs 8 and 12) [23], there is a need to study the effects of recycled materials as modifiers in composites with low toughness. With the use of recycled polymers or/and waste wood to make WPCs [24,25,26,27,28], and the studies presented in this work, companies can design impact-modified WPCs based on recycled materials, contributing to a more sustainable circular production approach. When considering materials that are currently disposed of in landfills or incinerated but could potentially be used as modifiers for enhancing toughness and impact properties, two types could be selected: recycled textiles (long fibers) and elastomers (soft inclusions). For example, every year, 12.6 million tons of textile waste is generated in Europe, approximately 77% of which goes to landfills or incinerators [29]. Textiles are woven from spun yarns, such as cotton or polycotton fibers, using an appropriate method, these recycled textiles can be converted back into fibers, turning them into excellent additives or reinforcements in composites. Another noteworthy factor is the estimated 185,000 tons of EPDM dumped into landfills every year [30]. As mentioned above, the use of EPDM elastomers has proven to be an excellent toughness modifier in WPC; therefore, recycled EPDM elastomers have the potential to provide similar behavior. These substantial quantities of waste textiles and elastomers can offer an opportunity to function as impact modifiers and thus contribute to the resource-efficient use of materials, promote recycling, and increase the circular economy.

Hence, this study focuses on WPCs with polypropylene as a matrix and wood fibers as reinforcement with the addition of recycled materials as impact modifiers. The recycled materials were denim fabric and recycled elastomer that were added targeting improved impact behavior and toughness reaching those of commercial WPCs modified with virgin fossil-based elastomer. The WPC microstructures were studied using scanning electron microscopy and X-ray microtomography. Mechanical characterizations, including impact tests, fracture toughness with digital image correlation, and tensile tests, were conducted.

## 2. Materials and Methods

### 2.1. Materials

WPC40: Commercially available WPC-grade DuraSense Pure 40 Food, with a melt flow index of 0.32 g/10 min (190 °C/2.16 kg), consisting of 60 wt% polypropylene (PP), 40 wt% wood fibers (WF), and a minor amount of maleic anhydride-grafted polypropylene (MAPP) was kindly supplied by Stora Enso (Hylte Mill, Hylte, Sweden), and it was used as base material for the compounding with the recycled materials.

Recycled-textile masterbatch (PP-rT): A denim fabric from used jeans (composition of 71% cotton, 27% polyester, and 2% elastane) was cut into 10–15-mm-wide strips, as shown in Figure 1a. Two different yarns were separated from the fabric shown in Figure 1b, the blue yarn is cotton fibers, and the white one is polyester fibers with thick elastane fibers as a minor component. The three separated fiber types are shown in a polarized optical microscopy image in Figure 1c. Struktol TPW 113 lubricant was purchased from the Struktol Company (St. Louis, MO, USA). Both the PP matrix and MAPP coupling agent were supplied by Stora Enso (Hylte Mill, Hylte, Sweden) and used for the preparation of the recycled textile masterbatch.

Recycled elastomer (rE): Consisting of 60 wt% PP and high-density polyethylene (PP-HDPE) and 40 wt% recycled EPDM elastomer with additives. It was kindly provided by Ecorub AB (Lövånger, Sweden), the rE pellets are shown in Figure 1d. According to the datasheet, the tensile modulus of rE is 0.34 GPa, strength 11 MPa, elongation at break 23%, and Charpy impact strength 21 kJ/m^2^.

WPC-E: A commercial impact-modified WPC, grade DuraSense Pure 30 Flex G, was used as a reference, consisting of 60 wt% PP, 30 wt% WF, 10 wt% elastomeric modifier, and a small amount of MAPP coupling agent. It was kindly supplied by Stora Enso (Hylte Mill, Hylte, Sweden).

Chemicals: Xylene purchased from Sigma-Aldrich (98%, Darmstadt, Germany) and ethanol 95% (Clean Chemical Sweden AB, Borlänge, Sweden) were used for Soxhlet extraction of the fibers from the matrix polymer for size measurements.

### 2.2. Processing

#### 2.2.1. PP-rT Masterbatch

A co-rotating twin screw extruder (TSE) (ZSK 18 MEGALab, Coperion GmbH, Stuttgart, Germany) was used to manufacture the PP-rT masterbatch, which consisted of 57.5 wt% PP, 40 wt% used denim fabric, and 2.5 wt% MAPP and lubricant combined. The coupling agent MAPP content was adjusted to be the same in the WPCs as in the WPC-E, and a small amount of lubricant was used to improve processability due to the high fiber content. PP, MAPP, and the lubricant were premixed and fed into the main inlet of the extruder using a K-SFS-24 gravimetric feeder (Coperion K-Tron GmbH, Stuttgart, Germany). The denim fabric was fed using the recycled textile long-fiber thermoplastic process (RT-LFT) [31]. The denim strip was fed into the side extruder, and the content was controlled by the side extruder screw speed, knowing the targeted fiber content, weight per meter of the fabric strip, and length per revolution of the screw used, according to Equation (1)
(1)$$\mathrm{Screw}\;\mathrm{speed}=\frac{\mathrm{targeted}\;\mathrm{fiber}\;\mathrm{content}}{\left(\mathrm{screw}\;\mathrm{length}/\mathrm{rev})\;\times \;\mathrm{time}\;\times \;(\mathrm{weight}/\;\mathrm{meter}\;\mathrm{denim}\;\mathrm{strip}\right)}$$

The total throughput was set to 4 kg/h and the targeted textile fiber content was 40 wt%, resulting in a feeding rate of 1.6 kg/h denim fabric. The length per revolution is 0.05 m, time 1 h (60 min), the weight of the fabric strip was 0.0044 kg/m, and the resulting speed was 122 rpm, which was set as the screw speed of the side extruder. A schematic of the PP-rT process is shown in Figure 2, where the temperature profile set between 180 and 195 °C is shown. The feeding rate of the polymer was 2.4 kg/h and the main screw speed for the extruder was 200 rpm. The extruded strands were cooled in a water bath, pelletized into granulates and dried overnight in an oven (60 °C) before further processing.

#### 2.2.2. Manufacturing of the Modified WPCs

WPCs with recycled modifiers were compounded using the same co-rotating TSE as used for the masterbatch production, described in the previous section. The textile masterbatch (PP-rT) was mixed with commercial WPC40 to obtain the final WPC-rT, and the recycled elastomer (rE) was mixed with WPC40 to obtain the final WPC-rE. The prepared WPCs had the same MAPP content as the commercially modified WPC (WPC-E). The materials are listed in Table 1.

For both WPCs, the processing parameters and screw design are shown in Figure 3, with a temperature profile between 180 and 200 °C, total throughput of 3 kg/h, and screw speed of 300 rpm. The materials were cooled in a water bath, pelletized to granulates using the strand pelletizer, and dried overnight in an oven (60 °C).

Samples for X-ray microtomography (µ-CT) and impact tests were produced using a compression-molding LabEcon 300 press (Fontijine Presses BV, Rotterdam, The Netherlands). The WPC granulates (15 g) were placed in a metal frame (130 × 130 × 1.5 mm^3^) between Mylar^®^ films (Lohmann Technologies, Milton Keynes, UK) and steel plates. The preheating time was 5 min at 200 °C, compression molding pressure was 5.9 MPa for 60 s, and cooling was conducted under the same pressure at 25 °C for 4 min. The samples for the µ-CT study were cut from the molded sheets using a CMA0604-B-A laser (Han’s Yueming Laser Group Co. Ltd., Dongguan City, China) to a size of 4 × 4 mm^2^.

For the fracture toughness test, the samples were injection molded into a rectangular shape (44 × 10 × 5 mm^3^), using a Haake II MiniJet Pro Piston Injection-Moulding System (JK Lab Instrument AB, Åkersberg, Sweden). The cylinder temperature was 200 °C, the mold temperature was 75 °C, the injection pressure was 400 bar and the holding time was 40 s, and the back pressure was 150 bar for 30 s. The single-edge-notch bending (SENB) specimen geometry is shown in Figure 4a, a notch with a depth of 3.5 (±0.5) mm was made with a saw with a blade thickness of 0.5 mm, and the pre-crack with a new microtome blade in the notch using compression mode in a test rig (Instron 4411 Series, Instron, Norwood, MA, USA) at a rate of 0.1 mm/min until a maximum displacement of −1.5 mm was reached (0.45 < a/W > 0.55), where a/W is the crack length across the width of the specimen. Both the notch and the pre-crack are shown in Figure 4b. All samples were painted white and speckled with a black dye to provide contrast for the digital image correlations (Figure 4c) and the test setup is shown in Figure 4d.

For tensile testing, WPC-rT, WPC-rE, and WPC-E granulates were injection-molded into a type-V (ASTM D638) [32] specimen, using the same Haake II MiniJet, with a cross-section area of 3.18 × 3.20 mm^2^. The cylinder temperature was 200 °C and the mold temperature was 75 °C, the injection pressure was 400 bar, the hold time was 40 s, followed by back pressure of 150 bar for 30 s.

### 2.3. Characterizations

#### 2.3.1. Microstructure of Raw Materials

A stereomicroscope (SM) (Nikon SMZ1270, BergmanLabora, Danderyd, Sweden) was used to study the yarns separated from the denim fabric, and single fibers separated from the textile were studied using a polarized optical microscope (POM) (Nikon Eclipse LV100POL, BergmanLabora, Danderyd, Sweden).

#### 2.3.2. Microstructure of Composites

The injection-molded specimens were fractured in liquid nitrogen, and the fracture surfaces were analyzed using a scanning electron microscope (SEM) (JSM-IT300LV, JEOL Ltd., Tokyo, Japan) at an acceleration voltage of 10 kV. The specimen surface was sputter-coated (Leica EM ACE200, Leica Microsystems, Wetzlar, Germany) with a 10-nm-thick layer of platinum to avoid charging.

X-ray microtomography (µ-CT) (Zeiss Xradia 510 Versa, Carl Zeiss, Pleasanton, CA, USA) was used to study the microstructure of the composite materials and dispersion of the textile fibers. Specimens were scanned using a 20× objective, which performed interior tomography with a field of view of 0.56 mm and voxel size of 0.56 μm to visualize the 3D structure of the materials. The scanned region of interest was positioned at the center of each sample and scanning was performed with an X-ray tube voltage of 50 kV, output power of 4 W, and without any X-ray filters. Reconstruction was performed using filtered back-projection with Zeiss Scout-and-Scan Reconstructor software (version 11.1), and 3D visualization and analysis of the structure were performed using Dragonfly Pro Software version 2022.2.0.1409 (Object Research Systems ORS Inc., Montreal, QC, Canada).

Measurement of fiber dimensions, the textile- and wood fibers were extracted from the composite (WPC-rT) pellets by boiling them in xylene at 170 °C for 2 h, followed by Soxhlet extraction in xylene (185 °C) for 24 h. Extracted fibers were filtered and washed with ethanol and distilled water and dispersed in water (0.1 wt%) using magnetic stirring. Textile fiber dimensions were measured using optical microscopy (OM) (Eclipse LV100PO, BergmanLabora, Danderyd, Sweden) and an image analysis program (ImageJ, 1.54d National Institutes of Health, University of Wisconsin, Madison, WI, USA). At least 100 fibers were measured.

#### 2.3.3. Falling Weight Impact Testing and Fracture Toughness

An Instron Dynatup Minitower Drop Weight Impact Testing device (Instron, High Wycombe, UK) was used for the impact load and energy measurements, and the sample was clamped onto a round sample holder (diameter 75 mm). The test was performed at 1.6 m/s and with 3.6 kg drop mass, resulting in 4.7 J impact energy. The composite materials were conditioned in RH 50% at 25 °C for 48 h prior to the tests. Images of the impact specimens were captured, and the damaged areas were measured using ImageJ software (National Institutes of Health, University of Wisconsin, WI, USA).

The fracture toughness tests were made following ASTM D5045 [33] standard in the bending mode (with a span length of 40 mm) for samples with sharp pre-cracks using an Instron 4411 Series (Instron, Norwood, MA, USA), (Figure 4d), with a crosshead speed of 10 mm/min and load cell of 500 N. Three to four samples were measured, and the average was calculated. Crack propagation was monitored using a JAI GO-5000M camera (Stemmer Imaging AB, Stockholm, Sweden) connected to a digital image correlation (DIC) system ZEISS ARAMIS GOM Correlate Software, version 2.0.1 (Carl Zeiss GOM Metrology GmbH, Braunschweig, Germany). After the test, crack propagation and length were analyzed using a stereomicroscope (Nikon SMZ1270, BergmanLabora, Danderyd, Sweden). Fracture surfaces along the x–y and y–z planes were also analyzed using SEM (JSM-IT300LV, JEOL Ltd., Tokyo, Japan) at an acceleration voltage of 10 kV, and the specimens were sputter-coated (Leica EM ACE200, Leica Microsystems, Wetzlar, Germany) with platinum (10-nm-thick layer). The critical-stress-intensity factor ($K_{\mathrm{IC}}$) for the SENB specimens was calculated following, Equations (2)–(4)
(2)$$K_{Q}=\left(\frac{P_{Q}}{B\sqrt{W}}\right)f(x)$$
(3)x=a/W

$P_{Q}$ is the determined load in the load-displacement diagram, B is the thickness, W is the width of the specimen, a is the crack length, and f(x) is defined as
(4)$$f(x)=6x^{1/2}\frac{\left[1.99-x(1-x)(2.15-3.93x+2.7x^{2}\right]}{(1+2x){\left(1-x\right)}^{3/2}}$$

This standard assumes linear elastic behavior of the specimen, and plane-strain at the crack tip, which can be validated by two conditions: Condition 1: The test is valid only if Equation (5) is satisfied, as follows
(5)$$P_{MAX}/P_{Q}<1.1$$
where $P_{MAX}$ is the maximum load that the specimen was able to sustain during the test.

Condition 2: The $K_{Q}$ is only valid if the size criteria (Equation (6)), which ensure plane strain fracture toughness, is satisfied if
(6)$$B,\;a,\left(W-a\right)\;>2.5\;{{(K}_{Q}/\sigma_{y})}^{2}$$
where $\sigma_{y}$ is the yield stress of the material, which was taken from the maximum stress obtained by the tensile test. If both conditions are satisfied that $K_{Q}$ is equal to $K_{\mathrm{IC}}$.

The strain energy release rate, $G_{\mathrm{IC}}$*,* at fracture initiation was calculated using Equation (7)
(7)$$G_{\mathrm{IC}}=\frac{K_{\mathrm{IC}}^{2}}{E}\times (1-v^{2})$$
where E is the modulus of the WPCs from tensile test conducted at the same strain rate as the fracture toughness and v is the Poisson’s ratio of the WPCs, calculated using Equation (8)
(8)$$v=v_{f}\times V_{f}+v_{m}\times (1-V_{f})$$
where $v_{f}$ is the Poisson’s ratio of the fiber, estimated to be 0.38; $v_{m}$ is the Poisson’s ratio of the matrix, estimated to be 0.42; and $V_{f}$ is the fiber volume fracture [20].

One-way analysis of variance (ANOVA) and Tukey’s honesty significant difference tests, with a significance level of 5%, were used for the results from the impact test, and fracture toughness tests using Past4 software 4.12b (Natural History Museum, Oslo, Norway).

#### 2.3.4. Mechanical Testing

The mechanical testing was performed using a universal testing machine (Autograph AG-X, JK Lab Instrument AB, Åkersberg, Sweden). Testing was performed according to the ASTM D638 standard, with a load cell of 5 kN, 25 mm between the grips, and a strain rate of 2.5 mm/min. A noncontact video extensometer model DVE-201 (JK Lab Instrument AB, Åkersberg, Sweden) was used to measure strain for at least five specimens. Samples were conditioned in RH 50% at 25 °C for 48 h prior to testing. In addition, the fracture surfaces from the tensile test were analyzed using SEM (JSM-IT300LV) at an acceleration voltage of 10 kV. The specimen surface was sputter-coated (Leica EM ACE200) with a 10 nm thick layer of platinum. The E modulus for $G_{\mathrm{IC}}$, was calculated from the tensile test with a crosshead speed of 10 mm/min (ASTM D5045) using an AGX-V series testing machine (BergmanLabora, Danderyd, Sweden) equipped with a TRViewX digital video extensometer (BergmanLabora, Danderyd, Sweden). ANOVA and Tukey’s honesty significant difference in both tests, with a significance level of 5%, was used for the results from the tensile test using Past4 software 4.12b.

#### 2.3.5. Eco-Audit Analysis Tool

The eco-audit tool in Granta EduPack (Level 3 Sustainability, Version 22.2.2, 2022, ANSYS, Canonsburg, PA, USA) was used for a simplified analysis of the environmental impact. In this analysis, the individual materials are analyzed in terms of their mass, and both energy and CO_2_ footprint only take into consideration virgin materials. In contrast, the values for recycled materials are zero.

## 3. Results and Discussion

The results of the different characterizations performed on the WPCs are discussed in several subsections. First, a section of the microstructure and morphology was used to understand how the different constituents (polymer, wood fibers, and modifiers) were dispersed and distributed within the composites. Secondly, results were reported from the different mechanical characterizations, mainly focusing on impact, fracture toughness, and toughness properties. This study focused on the effect of recycled materials as modifiers in the replacement of fossil-based modifiers presented in the commercially modified WPC, where comparable properties should be achieved for all WPCs.

### 3.1. Microstructural Analysis of the Composites

Figure 5 shows cross-sections of the WPCs, fractured in liquid nitrogen. Figure 5a shows that the denim fabric is separated into long and thin fibers, it is also possible to see some holes in the matrix from fiber pullouts, in addition, some thicker fibers that may be wood fibers are visible. Figure 5b shows the thin fiber at higher magnification, the surface is smooth and clean indicating that the adhesion with the polymer has been poor. One reason for the poor interaction may be the dye present in the recycled textile. Serra et al. [34] showed in a study that the presence of the dye in recycled cotton reduced the interaction between polypropylene matrix and cotton fibers. The cross-section of WPC-rE is shown in Figure 5c, it is difficult to distinguish the wood fibers from the fracture surface, this could indicate good adhesion between the wood fibers and the polymer or that the wood fibers are covered by the elastomer and therefore not clearly visible. Figure 5d a higher magnification shows round elastomeric particles in the matrix, also some holes that may be places where wood fibers have been before the break, and some wood fibers coated by rE. The fracture surface of WPC-E is shown in Figure 5e, some wood fibers can be distinguished, but these are broken, the fracture has occurred in wood fibers and no fiber pull-outs are visible, this indicates very good adhesion between the matrix and the wood fibers. Figure 5f shows a magnified view of broken wood fibers. Furthermore, the elastomeric phase is not visible in WPC-E, indicating good compatibility between the matrix and the elastomer.

Three-dimensional reconstructions from µ-CT are shown in Figure 6 (videos and higher magnifications are available in Appendix A), where different components of the composites are seen. The fibers are shown with a purple color as well as with a natural color. As expected, WPC-rT contains more fibers than the other WPCs, and it is possible to distinguish fine and long fibers that are separated from the denim fabric and no remnants of the fabric are visible. If comparing the WPC-rE and WPC-E, fewer fibers are visible in WPC-rE; however, there is a more green color which is the elastomeric modifier, this covers the fibers as discussed in the previous section and therefore a lower number of purple fibers are visible. Larger spots of purple particles/fibers are also visible, these are wood particles/fibers. The modifier in the WPC-E is more homogeneously distributed in the composite and has a smaller size compared with rE, and it appeared to have a better interaction with the matrix than with the fibers. Higher magnification of the 2D reconstruction is shown in Appendix A, where the recycled elastomer covers the fibers, whereas the commercial elastomer is evenly dispersed in the matrix.

Soxhlet extraction of the WPC-rT compounded material showed that the recycled textile fiber length was considerably reduced during the process, as shown in Appendix A. Recycled denim textile fibers are longer and have a smaller diameter than wood fibers; their length ranges from 65 to 2500 µm, with most in the range of 100–1100 µm. The aspect ratio for recycled denim textile fibers was found to be in the range of 5–130, with the majority ranging between 10 and 50. A previous study [35] reported that the average aspect ratio of the same wood fibers was lower than that of the rTF. Considering that previous studies used much longer fibers to provide toughness [19,20], rTFs are longer than wood fibers and may act as modifiers for WPCs via fiber pull-out. Further investigations will be conducted based on the fracture-toughness results. Regarding rE, it was assumed that the rE particle size would be close to that reported by Oksman and Clemons [1], ranging between 0.1 and 1 µm, with an average size of 0.3 µm.

### 3.2. Impact Strength and Fracture Toughness

Typical load–time and energy–time graphs for the drop-weight impact tests are presented in Figure 7. The WPC-rT presented the highest load-time as well as energy-time graphs, the higher load needed to initiate cracks while the WPC-rE showed the lowest load and energy-time. The WPC-E is performing between these two composites. All samples were completely perforated with the impactor. The weak interface between textile fibers and WPC and the length of the textile fibers resulted in a fiber pull-out mechanism while the addition of elastomers acted as stress concentrations, improving impact properties.

The WPCs exhibited similar impact behavior, including disturbance under the load, which is associated with crack initiation in the first milliseconds and the subsequent increase in load. The abrupt drop in the load was due to the total perforation of the impactor and complete breakage of the specimen (indicated by the arrows in the graph). The maximum impact load for WPC-rT was approximately 129 ± 3 N, the WPC-rE load was slightly lower (120 ± 6 N), and the WPC-E load was approximately 125 ± 5 N. The energy-time shows the energy at the peak load, which was found similar for all materials 1.18–1.20 J. The absorbed energy was calculated by integrating the load-deflection curves, as shown in Appendix A, where a higher average impact energy absorbed by the materials was found for WPC-rE, with a value of 1.5 (±0.3) J, followed by 1.3 (±0.2) J for WPC-E and 1.2 (±0.3) J for WPC-rT. However, the differences were small and not statistically significant according to ANOVA (5%) and Tukey’s tests, indicating that the energy absorption was similar for the WPCs. As mentioned previously, all specimens were broken during the impact test in the presence of fragments owing to the total perforation of the impactor. In addition, for WPC-rT and WPC-E, the area around the crack became white, which can be attributed to the crazing and cracking of the matrix, as also reported by Puech et al. [36]. However, it was not possible to observe a similar color change in WPC-rE, possibly because of the black color of the specimen (Appendix A). The calculated average area of damage was higher for WPC-rT (1715 mm^2^) and lower for WPC-E (1351 mm^2^); however, the values among all composites were not statistically different.

Table 2 shows the plane-strain fracture toughness ($K_{\mathrm{IC}}$) and energy release rate ($G_{\mathrm{IC}}$) of the WPCs, and the most representative load-displacement curves are shown in Figure 8, together with the DIC images. All materials exhibited linear behavior with a sudden load drop after the maximum load, indicating unstable crack growth [37] (represented by events 1 and 2 in Figure 8), and Appendix A. For both WPC-rT and WPC-E, the cracks exhibited a small deviation, as seen in the DIC images after the test, until complete failure was reached. For WPC-rE, the crack propagated longitudinal to the applied stress.

The average total time for the crack to propagation until complete for WPC-E was 31 s, followed by WPC-rE at 25 s, and WPC-rT at 17 s. The K_IC_ values were the same for WPC-rT and WPC-E, and a slight decrease was observed for WPC-rE. Even though the crack propagation had no deviation and a lower K_IC_ was found for WPC-rE, the presence of the modifier did not lead to a catastrophic and fast failure. The energy ($G_{\mathrm{IC}}$) was found similar for all WPCs made from recycled materials, while WPC-E presented higher energy.

By observing the fracture specimens along the crack propagation in SEM and stereomicroscope, the suggested fracture toughness mechanisms of the modifiers used in this study are shown in Figure 9, Figure 10 and Figure 11. For WPC-rT, long textile fibers play an important role, the energy is mainly dissipated by debonding, fiber pull-out, and bridging mechanisms (Figure 9), as suggested elsewhere [21]. The fiber pull-out will occur if the fibers are long, facilitating dissipation of energy along their length [21], and if the interaction between these fibers and the matrix is poor, as shown in Appendix A. In addition, the crack path can change when the crack hits the fibers [38]. A few microcracks were also observed in the specimens perpendicular to the crack path (Appendix A), in addition to the white lines observed in the stereomicroscope, which could be attributed to stress-whitening zones, as also observed in the impact specimens.

Based on the microstructure of the WPC-rE in Figure 10a,b, it is seen that the interaction is very strong between the recycled elastomer and wood fibers, in addition, the crack is propagating along the applied stress in the matrix. Few microcracks are visible, and the applied stress forms micro voids perpendicular to the crack at points with high-stress concentrations creating crazing in the polymer, and due to the continuous crack propagation. Those elongated fibrils were clearly seen in WPC-rE, but not in WPC-rT. The rE is present in large size around the harder wood fibers, aiding in the schematic depiction in Figure 10c. The main mechanism for WPC-rE is the soft elastomer between the fibers and matrix (Appendix A), this is improving the impact and toughness properties [1] while reducing the mechanical properties compared to WPC-E. As seen in SEM images and µ-CT the recycled elastomer was located near the wood fibers or their interface, with the formation of micro voids and crazing being the main toughening mechanism.

As discussed before, the commercial elastomer was homogeneously distributed in the PP matrix in a single phase, because of the good interaction between the polymer and the elastomer, and the fracture mechanism could be explained by the presence of microcracks perpendicular to the crack propagation (see Figure 11a,b). Broken fibrils and microcracks (Appendix A) are also observed in WPC-E, which could be the result of crazing. Additionally, crack deflection was observed for WPC-E, this mechanism aided in reducing the stress intensity at the crack tip [15]. Moreover, by performing a detailed examination using a stereomicroscope, it became apparent that stress-whitened zones developed along the crack, which may have been associated with craze-like features [18]. In summary, the fracture mechanisms could be observed as microcrack formation and failed fibrils due to crazing during crack propagation (Figure 11c).

### 3.3. Tensile Properties

Figure 12 shows the stress–strain curves of the WPCs and a summary of the mechanical properties is shown in Appendix A. It is seen that the WPC-rT has the highest strength and modulus followed by WPC-E and WPC-rE. The tensile strength of WPC-rT improved from 39 (WPC-E) to 46 MPa, and the tensile modulus, from 2.4 to 3.7 GPa, the improvement is because overall fiber content increased, from 30 to 40 wt% and the absence of soft elastomer. The average strain of 3.4% is slightly lower than the WPC-E 3.7% but the difference is not significant (seen in Appendix A). The tensile strength (30 MPa) and modulus (2.2 GPa) of WPC-rE are lower compared to WPC-E; the reason is that the soft recycled elastomer is, at least partially, on the wood fiber surfaces, reducing the strength and modulus. A soft interface can reduce the effectiveness of stress transfer from the matrix to the fibers [1]. The elastomeric encapsulation of the wood fibers slightly improved the elongation at break (to 4.1%), but it is not a significant improvement (Appendix A). This was previously discussed by Oksman and Clemons [1], where the elastomer-covered wood particle surfaces were more effective as toughness modifiers for WPCs but this reduced both strength and modulus more than an elastomer present as a soft phase in the matrix polymer. Fracture surface images of samples from tensile testing, are seen in Appendix A shows several places where the denim fibers have been pulled out from the matrix, it is difficult to see the wood fibers, which indicates better interaction between the wood fibers and polymer matrix. Appendix A shows WPC-rE, it is difficult to see the wood fibers and that indicates good adhesion between the fibers and the matrix, therefore it is suggested that the wood fibers are covered by the rE. The toughness, indicated by the area under the stress–strain curves, showed similar values for all WPCs of approximately 1.0–1.1 MJ m^−3^ (shown in Table 2), with no statistically significant difference.

### 3.4. Eco-Audit Analysis 

The eco-audit showed that WPC-E needs 58 MJ/kg and the WPCs with recycled modifiers reduced it to 46 MJ/kg. This shows that using only 10% recycled materials will have a positive effect on energy by 26%. The WPC-E had a CO_2_ footprint of 2.4 kg/kg material this was reduced to 1.8 kg/kg with recycled modifiers. It is important to point out that only the contribution of the virgin elastomer impacts 23% of the total CO_2_ footprint. The details of the analysis are shown in Appendix A.

## 4. Conclusions

This study aimed to assess the potential of recycled textiles and recycled elastomeric polymers in improving the properties of PP-based WPCs, especially impact and (fracture) toughness properties, to be on a similar level with the commercial impact-modified WPCs (WPC-E). The developed WPCs consisted of 60 wt% polypropylene, 30 wt% wood fibers, 10 wt% recycled impact modifiers, and a small amount of compatibilizer, manufactured using a co-rotating twin-screw extruder.

The microtomography results showed the recycled denim textile strip was separated into fibers that were well dispersed and distributed within the WPC. The manufactured WPCs did not show significant differences in terms of their impact properties and toughness. The composites with recycled denim fabric as a modifier (WPC-rT) exhibited fracture toughness like that of WPC-E, whereas the composite with recycled elastomer (WPC-rE) exhibited slightly lower fracture toughness. The toughening mechanism investigated in this study showed that the recycled denim fabric, which was successfully separated into fibers during the compounding, lacking interaction with the matrix, led to a fiber pull-out mechanism maintaining the fracture toughness. For WPC-rE, the recycled elastomer covered the wood fibers, suggesting a good interface between the fiber and the matrix, but with decreased strength and stiffness, and increased failure at break. In WPC-E, the elastomer was well-distributed in the polymer matrix, and the main toughening mechanisms were crazing and microcracks formation, with cracks slightly deviating from the original path.

Understanding the behavior of recycled materials is important for designing sustainable materials. In this study, the recycled denim fabric and elastomer were used as impact modifiers in WPCs. These modifiers especially the denim fabric resulted in impact properties and toughness like the commercial WPC but with better mechanical properties. This promotes the circular economy—where used materials are reintroduced into the economy, instead of being landfilled—reducing waste generation and better use of resources, and these findings could potentially lead to new WPCs based on 100% recycled materials.

## Figures and Tables

**Figure 1 polymers-16-02549-f001:**
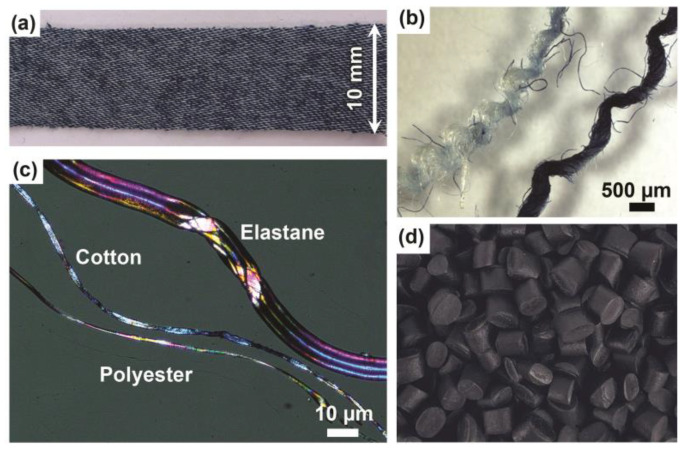
Used recycled materials in this study: (**a**) a strip of denim fabric; (**b**) two different types of yarns separated from the denim fabric, where the dark blue is dyed cotton and the white is polyester/elastane yarn; (**c**) polarized optical microscopy image showing the size difference of elastane, cotton, and polyester fibers separated from the yarns; and (**d**) image of the received recycled elastomer in granulated form.

**Figure 2 polymers-16-02549-f002:**
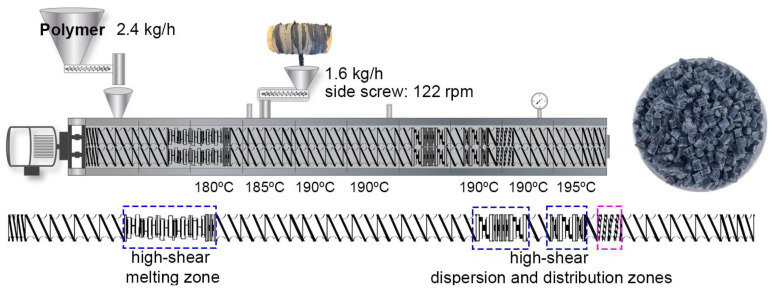
A schematic of the processing of textile masterbatch where the denim fabric was fed using a side extruder, the PP-rT masterbatch granulates, and the screw-configuration.

**Figure 3 polymers-16-02549-f003:**
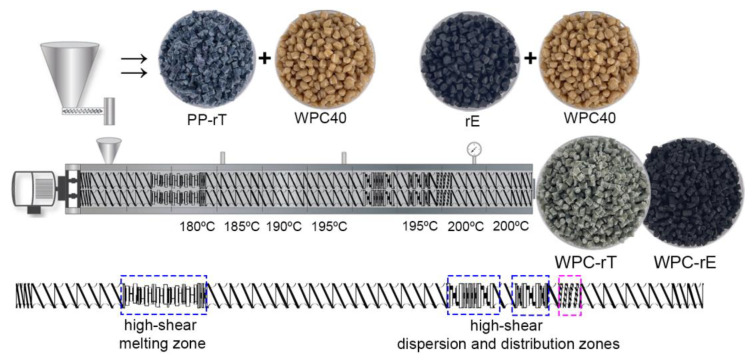
Extrusion profile temperatures at different zones. Granulated PP-rT was compounded with WPC40 to produce WPC-rT, and rE was also compounded with WPC40 to form WPC-rE. The screw profile with different elements is shown on the bottom of the picture, with the high-shear zones highlighted.

**Figure 4 polymers-16-02549-f004:**
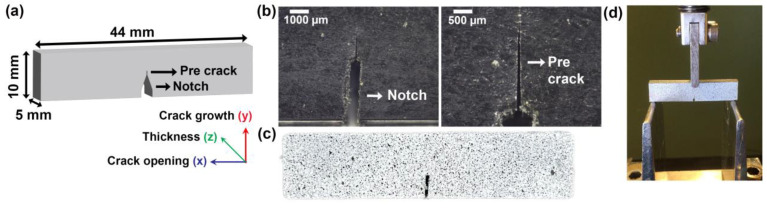
Fracture toughness test: (**a**) schematic of the used specimen’s shape; (**b**) stereomicroscope image of the notch and pre-crack; (**c**) specimen with speckle pattern for contrasting; and (**d**) fracture toughness test set-up in bending mode.

**Figure 5 polymers-16-02549-f005:**
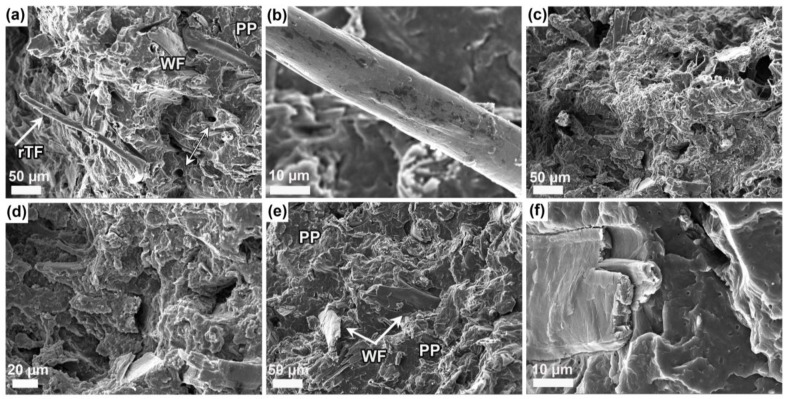
Fracture cross-sections of the WPCs: (**a**) WPC-rT; (**b**) higher magnification of the rTF with a clean surface; (**c**) WPC-rE; (**d**) magnified view of recycled elastomer (rE) covering the wood fibers; (**e**) WPC-E with some broken wood fibers; and (**f**) magnified broken wood fiber.

**Figure 6 polymers-16-02549-f006:**
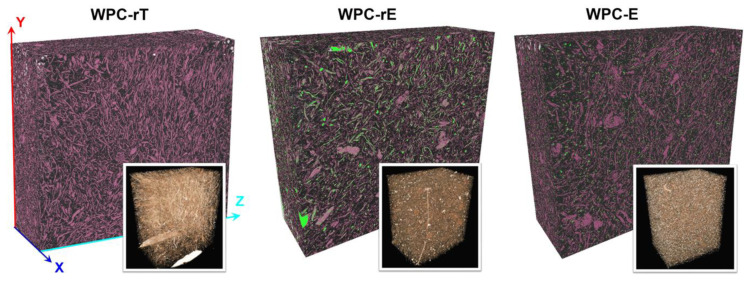
Three-dimensional reconstructions µ-CT of WPCs. Different colors represent different components in the composites. PP-matrix is black, fibers are purple, and elastomers are green; and bottom right figures show the same WPCs which are taken from the videos (available in Appendix A).

**Figure 7 polymers-16-02549-f007:**
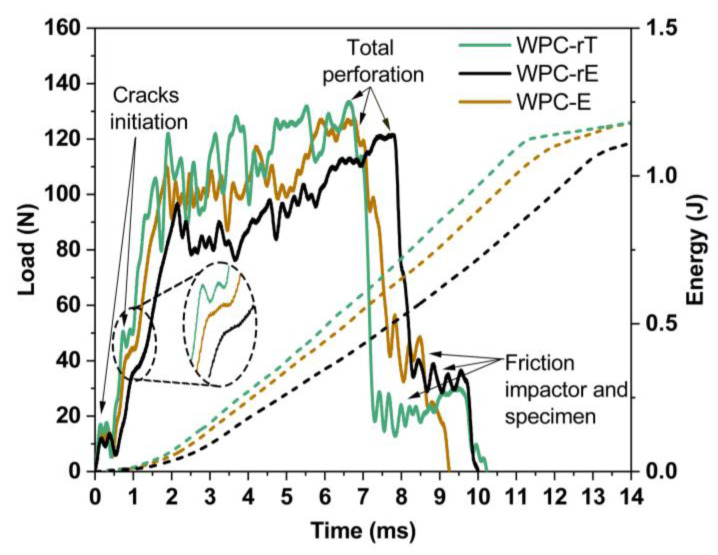
Load–time (continuous lines) and energy–time (dotted lines) curves from the drop-weight impact test, where crack initiation phases, total perforation of the impactor, and friction between impact and specimen are indicated with arrows.

**Figure 8 polymers-16-02549-f008:**
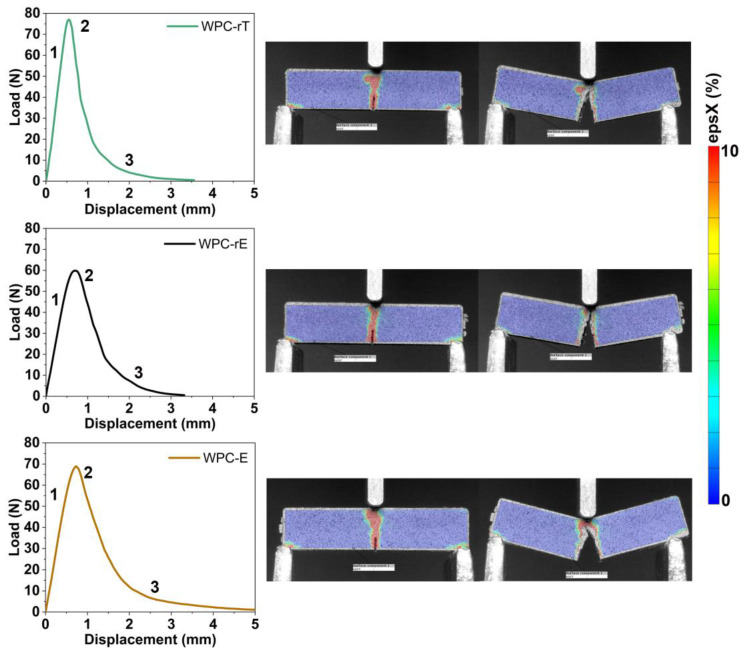
Load–displacement curves from fracture toughness with their respective DIC images (at the beginning of the test, and the end of the test); the color bar in DIC images represents the strain on the *x*-axis.

**Figure 9 polymers-16-02549-f009:**
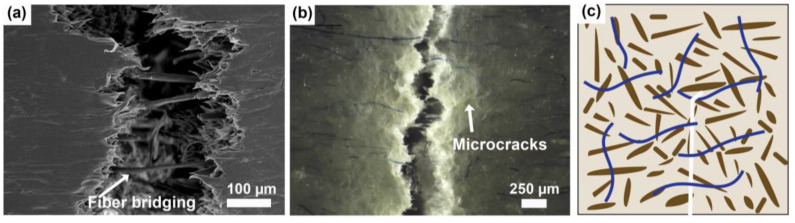
WPC-rT: (**a**) micrograph of a detailed view of crack from fracture toughness specimen, where fiber pull-outs and fiber bridging are visible; (**b**) stereomicroscope image of the crack, where fiber pull-outs and microcracks are also visible, as well as stress-whitening zone around the crack; (**c**) fiber pullout and fiber bridging are suggested fracture toughness mechanisms. Textile fibers are visualized as blue and wood fibers as brown.

**Figure 10 polymers-16-02549-f010:**
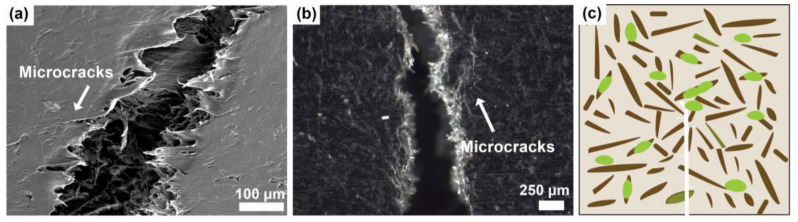
WPC-rE: (**a**) micrograph of the crack from the fracture toughness test showing crack propagation in the matrix with crazing mechanism; (**b**) a stereo-microscopy image of the crack; and crazing; and (**c**) the suggested fracture toughness mechanism. Wood fibers are visualized as brown and recycled elastomer as green.

**Figure 11 polymers-16-02549-f011:**
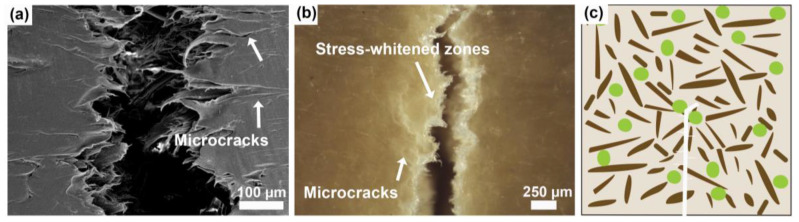
WPC-E: (**a**) micrograph of a detailed view of a crack from fracture toughness test where few microcracks are visible; (**b**) stereomicroscope image of the crack where microcracks are visible, along with a stress-whitened area; and (**c**) suggested fracture toughness mechanism. Wood fibers are visualized as brown and elastomer as green.

**Figure 12 polymers-16-02549-f012:**
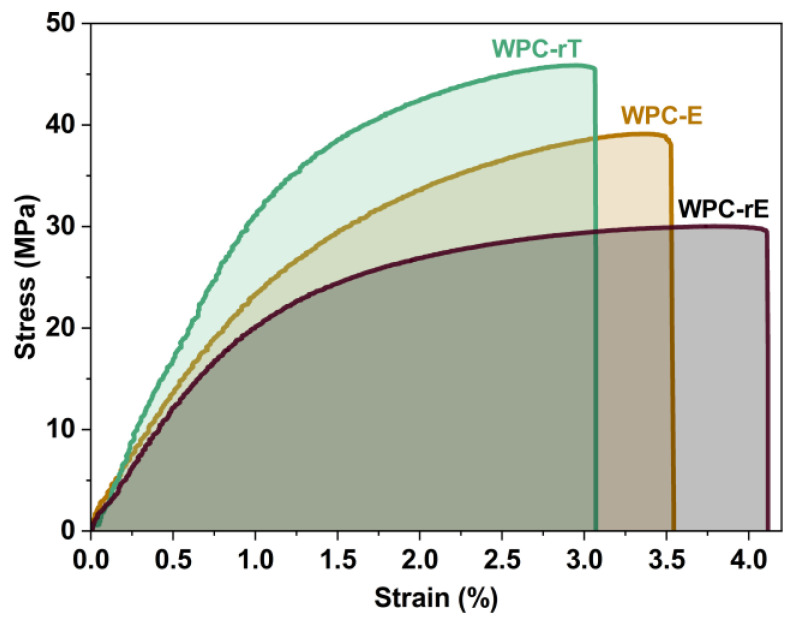
Representative stress–strain curves from tensile testing, where the area under the graph represents the toughness.

**Table 1 polymers-16-02549-t001:** Coding and material compositions, commercial grade of WPC40, manufactured masterbatch PP-rT, recycled textile-modified WPC (WPC-rT), recycled elastomer-modified WPC (WPC-rE), and commercial impact-modified elastomer WPC-E.

Material Codes	Polymer (wt%)	WF (wt%)	Impact Modifier (wt%)	Processing Aids (wt%)
WPC40	60	40	-	-
PP-rT	57.5	-	40	2.5
WPC-rT	60	30	10	-
WPC-rE	60	30	10	-
WPC-E	60	30	10	-

**Table 2 polymers-16-02549-t002:** Fracture toughness ($K_{\mathrm{IC}}$), energy ($G_{\mathrm{IC}}$), and toughness for the WPCs. (Marked with the same letter within the same column are not significantly different at a 5% significant level based on ANOVA and Tukey’s test).

WPCs	*K_IC_* (MPa m^1/2^)	*G_IC_* (kJ m^−2^)	Toughness (MJ m^−3^)
WPC-rT	1.44 ± 0.08 ^A^	0.48 ± 0.1 ^A^	1.1 ± 0.1 ^A^
WPC-rE	1.13 ± 0.10 ^B^	0.45 ± 0.1 ^A^	1.1 ± 0.2 ^A^
WPC-E	1.44 ± 0.09 ^A^	0.73 ± 0.1 ^B^	1.0 ± 0.1 ^A^

## Data Availability

The original contributions presented in the study are included in the article/Appendix A, further inquiries can be directed to the corresponding author.

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
