# Peer review of "Use of Recycled Additive Materials to Promote Efficient Use of Resources While Acting as an Effective Toughness Modifier of Wood–Polymer Composites"

_polymers, 2024, doi:10.3390/polym16182549_

Round 1

Reviewer 1 Report

Comments and Suggestions for Authors

This manuscript entitled “Use of recycled additive materials to promote efficient use of resources while acting as an effective toughness modifier of wood-polymer composites” presents a timely and innovative approach to enhancing the properties of wood-polymer composites using recycled materials. The focus on sustainability and resource efficiency is particularly relevant in today’s environmental consciousness and material scarcity context.

I am pleased to recommend that the manuscript be accepted for publication in the Polymers journal, subject to minor revisions. As suggested below, specific sentences require correction.

Abstract: Line 16. Suggestion: “denim fabric into long, thin fibers that were well dispersed within the WPC, while the recycled elastomer was found close to the wood fibers, acting as a soft interphase.”

Introduction: Line 35. Suggestion: “Few studies have investigated the impact and fracture properties of WPCs [3,5], the use of modifiers [1,4,6–14], or instances where recycled materials were used as impact modifiers in WPCs are challenging to find.”

Materials and Methods: Line 123. Suggestion: “Recycled elastomer (rE): Consisting of 60 wt% PP and high-density polyethylene (PP-HDPE) and 40 wt% recycled EPDM elastomer with additives, as shown in Figure 1(d), was provided by Ecorub AB (Lövånger, Sweden).”

Processing: Line 142. Suggestion: “A co-rotating twin-screw extruder (TSE) (ZSK 18 MEGALab, Coperion GmbH, Stuttgart, Germany) was used to manufacture the PP-rT masterbatch, which consisted of 57.5 wt% PP, 40 wt% used denim fabric, MAPP, and lubricant (2.5 wt%).”

Results and Discussion: Line 338. Suggestion: “Figure 5(f) shows a magnified view of broken wood fibers.”

-------------------------additional comments-------------------------

1. What is the main question addressed by the research?

The manuscript addresses the significant need for more environmentally friendly building materials in the context of global resource shortages.

  2. Do you consider the topic original or relevant in the field? Does it address a specific gap in the field? What does it add to the subject area compared with other published material?
This research fills a critical gap by demonstrating practical applications of waste materials in new, value-added contexts.
3. What specific improvements should the authors consider regarding the methodology? What further controls should be considered?
The additional control (if any) could include repeated trials to test the consistency of the composite properties across different batches of recycled materials.
4. Illustrate what are, in your opinion, its strengths and weaknesses (this is an essential step, as the editor will consider the reasoning behind your recommendation and needs to understand it properly)
The study is pioneering in its approach and provides a clear, detailed analysis of the material properties.

5. Are the conclusions consistent with the evidence and arguments presented and do they address the main question posed?
The conclusions are well-supported by the data and analyses provided, directly addressing the main research question.

6. Are the references appropriate?
I think the references are appropriate and adequately support the manuscript.

7. Please include any additional comments on the tables and figures and quality of the data.
The tables and figures are well-prepared and effectively aid in understanding the experimental results.

Comments on the Quality of English Language

The manuscript is generally well-written but would benefit from minor editing to enhance its clarity and readability.

Reviewer 2 Report

Comments and Suggestions for Authors

This study presents a well-structured and comprehensive study on the use of recycled denim fabric and elastomer as modifiers to enhance the toughness of wood-polymer composites (WPCs). The authors have meticulously documented the materials and methods used, providing a clear pathway for reproducibility. The results are presented in a detailed manner, supported by thorough microstructural and mechanical characterizations, which highlight the potential of recycled materials to replace virgin fossil-based modifiers in commercial WPCs. The eco-audit analysis is a significant addition, demonstrating the environmental benefits of using recycled materials. Overall, the study makes a valuable contribution to the field of sustainable materials and circular economy.

Here are some questions below. I believe the manuscript can be accepted if the reviewers address these within the revised text (Major revision);

  1. The study mentions that the recycled denim fabric was successfully fibrillated into long and thin fibers. How does the interaction between these recycled fibers and the polymer matrix compare to the interaction with virgin fibers, especially in terms of bonding strength and durability over time?
  2. The study found that the length of the recycled denim fibers significantly varied after processing. How does the variation in fiber length impact the mechanical properties of the composites? Specifically, what is the threshold fiber length below which the toughness benefits diminish?
  3. The microtomography results indicate differences in the distribution of modifiers within the composites. Can you elaborate on how the homogeneity of the recycled modifiers' distribution affects the overall mechanical properties of the WPCs? Are there any specific processing adjustments that could improve this distribution?
  4. The fracture toughness tests indicated different failure mechanisms between the composites with recycled denim and those with recycled elastomer. Could you provide a more detailed analysis of these mechanisms? How do they compare to the failure mechanisms in composites modified with virgin materials?
  5. The fracture toughness (KIC) of WPCs with recycled denim fabric matched that of commercial WPCs, while the recycled elastomer showed slightly lower values. What factors do you believe contribute most significantly to these differences in performance? Is there a potential to further optimize the recycled elastomer formulation to match or exceed the toughness provided by virgin materials?
  6. The eco-audit analysis showed a reduction in energy use and CO2 footprint when using recycled modifiers. Could you provide more details on the specific methodologies and assumptions used in this eco-audit? How sensitive are these results to changes in the recycling process or the quality of the recycled materials?

Reviewer 3 Report

Comments and Suggestions for Authors

Dear Authors,

The work received for review is innovative and very interesting. It highlights the interest in recycled materials in the production of polypropylene-based WPC composites. I consider the description of the research detailed and appropriate. They emphasize the areas of WPC properties that are important from the point of view of the user and industry. I consider the topic of the work to be original. However, I would have a few minor comments:

1) The title of the work suggests the researchers' interest in a wider range of WPC materials. Meanwhile, the authors refer only to WPC manufactured on the basis of PP. Can the conclusions obtained also be applied to WPC composites based on, for example, PVC?

2) Why focus only on WPC composites based on polypropylene?

3) Figure 8 is a bit blurry. Maybe it should be enlarged or sharpened?
